# Skeletal Dysplasia: A Case Report

**DOI:** 10.3390/diagnostics13182905

**Published:** 2023-09-11

**Authors:** Nicolae Gică, Gabriela Mîrza, Corina Gică, Anca Maria Panaitescu, Anca Marina Ciobanu, Gheorghe Peltecu, Iulia Huluță

**Affiliations:** 1Gynecology Department, Faculty of Medicine, Carol Davila University of Medicine and Pharmacy, 020021 Bucharest, Romania; mat.corina@gmail.com (C.G.); anca.panaitescu@umfcd.ro (A.M.P.); anca.ciobanu@umfcd.ro (A.M.C.); gheorghe.peltecu@umfcd.ro (G.P.); iulia.huluta@drd.umfcd.ro (I.H.); 2Clinical Hospital of Obstetrics and Gynaecology Filantropia, 011171 Bucharest, Romania; gabrielamirza07@gmail.com

**Keywords:** skeletal dysplasia, hydrops, hypomineralization, micromelia

## Abstract

This paper presents a rare case of fetal hydrops detected at just 23 weeks of gestation in a 22-year-old woman’s first pregnancy. The fetal ultrasound revealed severe skeletal anomalies, craniofacial deformities, and thoracic abnormalities, suggesting a complex and severe skeletal dysplasia, potentially type IA Achondrogenesis—a lethal autosomal recessive condition marked by ossification delay. This case highlights the significance of advanced genetic testing, such as next-generation sequencing (NGS) and whole-genome sequencing (WGS), in diagnosing and understanding skeletal dysplasias. Skeletal dysplasias represent a group of genetic disorders that affect osteogenesis. The prevalence of this condition is 1 in 4000 births. Sadly, 25% of affected infants are stillborn, and around 30% do not survive the neonatal period. There is a wide range of rare skeletal dysplasias, each with its own specific recurrence risk, dysmorphic expression, and implications for neonatal survival and quality of life. When skeletal dysplasia is incidentally discovered during routine ultrasound screening in a pregnancy not known to be at risk of a specific syndrome, a systematic examination of the limbs, head, thorax, and spine is necessary to reach the correct diagnosis. Prenatal diagnosis of skeletal dysplasia is crucial for providing accurate counselling to future parents and facilitating the proper management of affected pregnancies. An accurate diagnosis can be a real challenge due to the wide spectrum of clinical presentations of skeletal dysplasia but advances in imaging technologies and molecular genetics have improved accuracy. Additionally, some of these skeletal dysplasias may present clinical overlap, making it especially difficult to distinguish. After the 11th revision of genetic skeletal disorder nosology, there are 771 entities associated with 552 gene mutations. The most common types of skeletal dysplasia are thanatophoric dysplasia, osteogenesis imperfect, achondroplasia, achondrogenesis, and asphyxiating thoracic dystrophy.

This is a rare case of a 22-year-old woman referred to our fetal department due to fetal hydrops detected at 23 weeks of gestation. This was her first pregnancy from a non-consanguineous marriage, and she had no significant medical history. The patient initially sought medical attention at 23 weeks of gestation, primarily due to social issues. The ultrasound examination in our department revealed multiple abnormalities in the fetus, prompting further investigation and management.

Skeletal abnormalities (see Figure 1A,B):Short limbs with severe micromelia affecting all four limbs.Remarkably bowed femur and humerus.Bilateral clubfoot.

Craniofacial abnormalities (see Figure 2A,B):Hypomineralization of the skull bones.Enlarged head and small face.

Thorax abnormalities (see Figure 3A,B):Narrow thorax with evidence of pulmonary hypoplasia.Multiple rib fractures.

Fetal hydrops (See Figure 2A and Figure 3A,B)Presence of fetal hydrops, characterized by ascites, pleural effusion, and subcutaneous edema.

The constellation of abnormalities observed in this case is highly indicative of complex and severe skeletal dysplasia. The prognosis for the fetus is concerning due to the severity of the abnormalities and associated complications. Further diagnostic investigations and genetic counselling were warranted to determine the underlying etiology and provide appropriate management and counselling for the patient and her family. The abnormalities observed were highly suggestive of type IA Achondrogenesis, which is a rare lethal autosomal recessive form of skeletal dysplasia characterized by a severe delay of ossification [1]. Diagnosis is based on integrated radiological, histological, and genetic findings. By enabling accurate diagnosis, subtype classification, risk assessment, accurate assessment of recurrence risks in subsequent pregnancies, and insights into pathogenesis, genetic testing plays a pivotal role in prenatal care and counselling. Molecular genetic testing focuses on identifying specific gene mutations and variants implicated in skeletal dysplasias. Techniques such as Sanger sequencing, next-generation sequencing (NGS), and targeted gene panels facilitate the detection of disease-causing mutations in genes associated with skeletal development. NGS platforms, including whole-exome sequencing (WES) and whole-genome sequencing (WGS), offer comprehensive genetic analysis, enabling the identification of novel mutations and a broader understanding of skeletal dysplasia etiology [2,3]. After birth, examination of skeletal radiographs is particularly important, as the classification of skeletal dysplasias heavily relies on radiographic findings [4].

**Figure 3 diagnostics-13-02905-f003:**
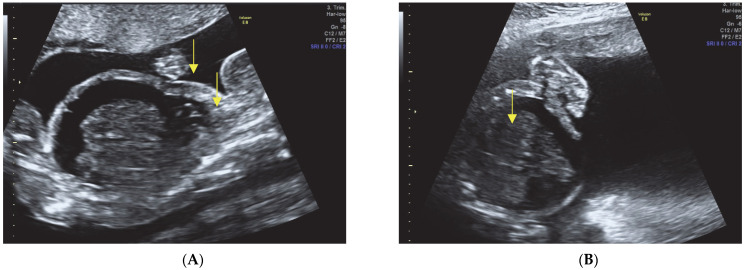
The narrow thorax, pulmonary hypoplasia, and multiple rib fractures are consistent with compromised lung development and potentially respiratory distress. This could be due to limited space within the thoracic cavity caused by severe skeletal abnormalities. These findings further contribute to the overall poor prognosis. (**A**) Midsagittal section at the level of the thorax and abdomen showing extremely narrow chest. Fetal hydrops, defined as abnormal fluid accumulation in two or more fetal compartments, is a significant finding associated with poor perinatal outcomes [5]; (**B**) Multiple rib fractures in the axial view of the fetal chest. Pleural effusion.

Unfortunately, despite our efforts this patient declined all further investigations and tests. She was offered the option of termination considering this is a lethal condition, and after discussing it with her partner she opted for termination on the same day (see Figure 4).

In summary, this case report gives us an insight into the complexity of the diagnosis of skeletal dysplasias. These conditions stem from pervasive disruptions in bone development that commence during fetal growth and persist throughout an individual’s lifespan. The prenatal diagnostic process for suspected skeletal dysplasia entails comprehensive ultrasound imaging of various anatomical areas, including the long bones, thoracic region, extremities, cranial area, spinal column, and pelvic region. Achieving a precise diagnosis is reliant upon genetic testing, as it plays a crucial role in accurately identifying and understanding these conditions.

## Figures and Tables

**Figure 1 diagnostics-13-02905-f001:**
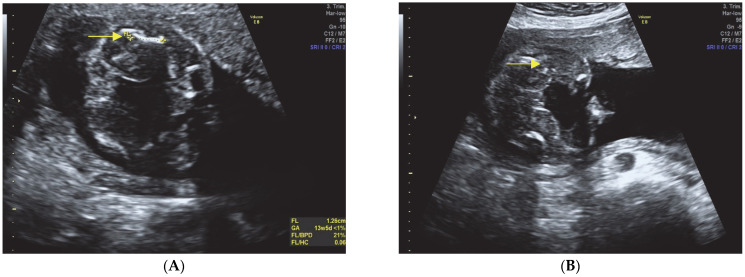
Assessing long bones on 2D ultrasound reveals signs of shortening, reduction, and extreme bowing, involving the entire limb (micromelia). (**A**) Fetal legs with remarkably bowed and short femur; (**B**) Fetal legs with bilateral clubfoot.

**Figure 2 diagnostics-13-02905-f002:**
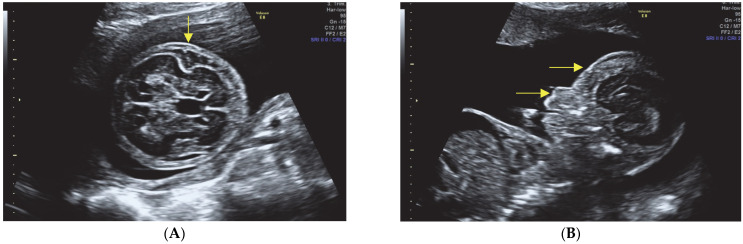
Reduced ossification of skull bones and facial abnormalities with a small face, absent nasal bone, and micrognathia. (**A**) Fetal head with hypomineralisation of the skull bones. Note the large and round head and the skin edema; (**B**) Fetal profile with a small face, absent nasal bone, and micrognathia.

**Figure 4 diagnostics-13-02905-f004:**
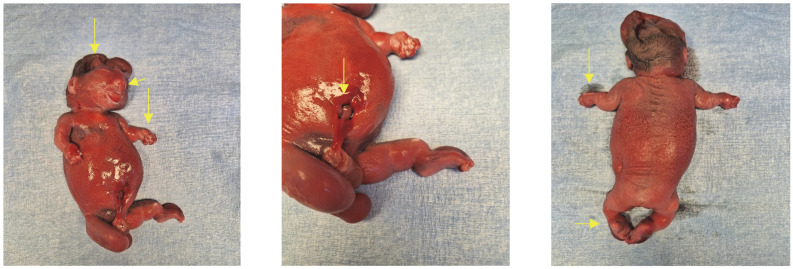
Patient phenotype shows a flat face, short nose, protruding tongue and eyes, low-set ears, narrow bell-shaped thorax, severe micromelia, bilateral clubfoot, and severe hypomineralisation of the calvaria.

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
