# Peer review of "Skeletal Dysplasia: A Case Report"

_diagnostics, 2023, doi:10.3390/diagnostics13182905_

Round 1

Reviewer 1 Report

The authors present the prenatal diagnosis of a case with skeletal dysplasia. Although this presentation is interesting, a paragraph dedicated to a condensed version of literature review should be added under the subheading "conclusion" This condensed version of literature review should focus on related sonographic findings during pregnancy and, thus, the references part should be enriched with newer and more up-to-date ones. I recommend that this manuscript can be accepted for publication in Diagnostics journal after this major correction has been made.

Author Response

Dear Reviewers,

Thank you for your suggestions. I am writing to submit the revision of the case report for consideration to be published in MDPI. The case report, titled "Skeletal Dysplasia- Case report" presents a unique and insightful clinical scenario that I believe would contribute to the medical literature in the field.

The abstract now incorporates the objective of the case report, and visual aids in the form of arrows have been thoughtfully included within the images. Additionally, a new subheading has been introduced. Under the subheading "Conclusion," a succinct summary of the prenatal diagnosis has been integrated. This adjustment has been made considering the nature of the article, as an exhaustive literature review could potentially be overwhelming for this type of article.

I have ensured that all necessary patient confidentiality and ethical considerations have been addressed in the manuscript.

Thank you for your time and consideration. I look forward to your response. Please feel free to contact me at gica.nicolae@umfcd.ro if you require any additional information or have any questions regarding the submission.

Sincerely,

Nicolae Gică

Reviewer 2 Report

The short case report of Huluta and colleagues is overall well written and of merit. I just have two minor comments:

- In the abstract I would specify the aims of the paper (presented now at the beginning of introduction)

- I would use arrows and/or symbols to illustrate inside the figure what the Figure description refers to

Author Response

(The authors gave the same response as above.)
